# Effect of Curing Agent on the Compressive Behavior at Elevated Test Temperature of Carbon Fiber-Reinforced Epoxy Composites

**DOI:** 10.3390/polym11060943

**Published:** 2019-05-31

**Authors:** Simon Bard, Martin Demleitner, Regino Weber, Rico Zeiler, Volker Altstädt

**Affiliations:** Department of Polymer Engineering, University of Bayreuth, Universitätsstr. 30, 95444 Bayreuth, Germany; Martin.Demleitner@uni-bayreuth.de (M.D.); Regino.Weber@uni-bayreuth.de (R.W.); Rico.Zeiler@uni-bayreuth.de (R.Z.); altstaedt@uni-bayreuth.de (V.A.)

**Keywords:** Celanese, elevated temperature, compression, carbon fiber, prepreg

## Abstract

The aim of the underlying research is to understand the effect of elevated test temperatures on the mechanical properties of carbon fiber-reinforced laminates based on three different hardeners. A high-temperature stable adhesive was developed for the end tabs of the specimen. Bifunctional bisphenol A diglycidyl ether (DGEBA) epoxy cured with triethylenetetramine (TETA), isophorone diamine (IPDA), and 4,4′-diaminodiphenylsulfone (DDS) were cured and tested in a Celanese compressive test up to 250 °C. A model by Feih was applied, and sufficient accordance (R2 > 97%) with the compressive data was found. We showed that the network density and the chemical structure of the thermoset network influenced the compressive behavior.

## 1. Introduction

Due to the increased complexity of lightweight design, polymer composites especially in aerospace and automotive applications are expected to withstand high temperatures. The evaluation and understanding of the properties of carbon fiber-reinforced polymers (CFRP) at elevated temperatures are therefore crucial to predict the behavior of these materials. This might be the case for structural parts of helicopters, such as tail booms, which can be exposed to high temperatures and heat from the engine for a short time [1]. Furthermore, in the automotive industry, parts in engine underhood applications close to the engine are expected to withstand fairly high temperatures [2].

Fiber-reinforced polymers (FRP) are widely used in transportation and construction due to their specific strength and stiffness allowing lightweight construction. Either thermoplastic or thermosetting matrix polymer composites are highly susceptible to temperature, which makes the understanding of their mechanical stability at high temperatures very important for structural composite design.

In the literature, research groups mainly use dynamic mechanical thermal analysis (DMTA) to evaluate the thermo-mechanical behavior of fiber-reinforced composites [3]. Although DMTA is useful to evaluate differences between different polymers and additives, it fails to provide useful data for the design of composite structures, since mostly mechanical behavior under torsion is observed for endless fiber-reinforced composites. Results from tensile, bending, and compressive tests are needed to accelerate the design optimization process of structures by numerical approaches [4].

Kandere et al. shed some light on the testing and simulation of the bending behavior of fiber-reinforced composites [5,6]. They elaborately studied the thermal and thermo-mechanical properties to develop kinetic, heat transfer, and macro-mechanical models in order to simulate the thermo-mechanical response of composites during heat/fire exposure. The researchers used an epoxy matrix reinforced by glass fibers and evaluated the thermo-mechanical behavior by thermo-gravimetric analysis and bending tests up to 300 °C. Hereby, their samples where heated by a fire source from one side. The thermo-mechanical model as applied in their study has previously been used to predict the time to failure for laminates simultaneously loaded and heated in fire.

So far, research is missing for compressive tests of CFRP at elevated temperatures. The difficulty lies in introducing the load into the specimen. Due to their low transverse and interlaminar strength, direct end loading is not suitable for the specimen [7]. Instead, shear-loading should be introduced via end tabs. Thereby, suitable adhesives are needed to ensure that no failure occurs between end tabs and specimen when the sample is heated up during the test. Consequently, we first developed a toughened adhesive with T_g_ > 300 °C to combine end tabs and specimen to ensure proper test conditions. 

In compressive tests, the matrix plays an important role in the failure behavior, as it prevents the fibers from buckling [6]. The influence of the resin becomes especially crucial for the mechanical properties under elevated temperatures. The thermo-mechanical properties of different hardeners and thereby different network densities have been studied in various studies by DMTA [8,9]. However, the mechanical behavior of laminates in compressive tests based on epoxy with different network densities at elevated temperatures has not been studied yet. An elaborative study about the influence of different network densities will lead to a deeper understanding of the influence of the matrix in mechanical testing, especially for compressive tests.

Hereby, the scientific motivation is to understand the effect of the matrix network chemistry on the compressive behavior of its carbon fiber-reinforced composites up to 250 °C. To achieve this main scientific aim, a standard bifunctional diglycidyl ether of bisphenol A epoxy resin was cured stoichiometrically with aliphatic, cycloaliphatic, and finally with an aromatic hardener, resulting in relatively low, medium, and high glass transition temperatures. Triethylenetetramine (TETA) is used for sport and transport applications, isophoronediamine (IPDA) for rotor blades or leaf springs, and 4,4′-diaminodiphenylsulfone (DDS) for aerospace applications. All laminates were tested above the glass transition temperature to determine the remaining strength at elevated temperatures. These values might be interesting when parts are exposed to short-term loads. Consequently, to be able to test the fiber-reinforced composite of these matrices up to 250 °C, a special end-tap adhesive which can withstand the shear loading at these temperatures had to be developed. This adhesive was a low temperature curing epoxy, preventing a major post curing of the composites during compressive sample preparation with end taps. Finally, the compressive testing of the composites with different epoxy matrices was performed. To deepen the understanding of the mechanical behavior of these materials, the testing results were evaluated together with the resin chemistry and network structure. 

## 2. Experiments

### 2.1. Materials

The resin systems consisted of a standard bisphenol A diglycidyl ether (DGEBA) (D.E.R. 331, Olin, Clayton, MO, USA) with an epoxide equivalent weight of 187 ± 5 g/eq. As mentioned before, three different hardeners have been used to cure the resin, which are triethylenetetramine (TETA) (REN HY 956, Huntsman, TX, USA), isophoronediamine (IPDA) (BASF Baxxodur TM EC 201, Ludwigshafen, Germany), and 4,4′-diaminodiphenylsulfone (DDS) (Organica Feinchemie GmbH, Wolfen, Germany). Table 1 shows the chemical structures and other relevant properties of the hardeners. The resin and hardener were mixed in stoichiometric ratios, for 100 g of the resin, 13.5 g of TETA, 42.58 g of IPDA, or 33.4 g of DDS were used.

A standard carbon fiber biaxial textile G0939 (Hexcel, Stamford, CT, USA) with an aerial weight of 220 g/m^2^ was chosen as endless fiber reinforcement. The textile is based on Toho Tenax High Tenacity 5131 3K polyacrylonitrile fibers with a tensile strength of 3950 MPa.

### 2.2. Productions of CFRP Laminates

Laminates were produced by a vacuum-assisted resin transfer molding (VARTM or RTM) with a 2K injection system (Isojet, Cordas, France) with 6 bar injection pressure. Laminates with dimensions of 400 × 400 mm featuring a [(90/0)]_4S_ (eight biaxial fabric layers in total) layup were manufactured for general purpose testing, leading to laminates with 2 mm thickness. Laminates with TETA and IPDA were produced by two-component injection, with the temperature of the resin tank, heating cover, and injection line set to 50 °C. The DGEBA/DDS resin system was pre-mixed for 30 min at 400 rpm and 110 °C with a laboratory mixer until the DDS hardener was dissolved in the resin. Subsequently, the reactive mixture of DGEBA/DDS was injected (1K) into the mold. The curing cycles for each type of system were set according to the manufacturer and are as shown in Table 2. Heating and cooling rates of 5 K/min were used. The curing cycles were confirmed by differential scanning calorimetry (DSC) measurements and dynamic mechanical thermal analysis (DMTA) measurements. In the DMTA, the sample was heated up twice. No shift in the maximum of loss tangent was detectable, which confirmed that the samples were fully cured.

### 2.3. Adhesive Development and Sample Preparation

It was necessary to develop an adhesive which withstands high temperatures to perform the compression tests. Commercially available adhesives, such as Elantech ADH 891.892 (3M, Maplewood, MN, USA), Uhu-Endfest (UHU GmbH & Co. KG, Bühl, Germany), and Scotch-weld DP760 (3M, Maplewood, MN, USA), which are sold as adhesives to withstand high temperatures, were tested in lap shear tests. Already at 200 °C, the commercially available adhesives lost more than 80% of their shear strength. The formulation for the developed adhesive consists of 41 wt% of an tris-(hydroxyl phenyl) methane-based epoxy (Tactix 742, Huntsman, TX, USA), 22 wt% triglycidyl-p-aminophenol (EPIKOTE resin 498, Hexion, Columbus, OH, USA), 12 wt% core-shell particles (ALBIDUR EP 2240 A, Evonik, Essen, Germany), and 25 wt% 4,4′-diaminodiphenyl sulfone (Organica Feinchemie GmbH, Wolfen, Germany). In the lap shear tests used for the commercial adhesives, the shear strength of the formulation decreased by 20%.

End tabs consisting of epoxy DuroBest250 and woven glass fiber (AGK Hochleistungswerkstoffe GmbH, Dortmund, Germany) were bonded to the sample with the adhesive mentioned above. It was cured at 180 °C for 60 min and post-cured for 5 min at 250 °C. The two-component adhesive Elantech ADH 891.892 (3M, Maplewood, MN, USA) was used for the samples tested up to 150 °C and cured for two hours at 80 °C. Samples were then cut with Diadisc 6200 (Mutronic, Rieden am Forggensee, Germany) to the dimensions specified in the standard. Strain gauges were not used as they failed at higher temperatures in prior tests.

### 2.4. Thermal Analysis

Thermo-gravimetrical analyses were conducted with Thermo-Gravimetry F1 Libra (Netzsch GmbH & Co. KG, Selb, Germany) between 25 and 400 °C under nitrogen atmosphere with an N_2_ flow of 50 mL/min and up to 250 °C under air with a heating rate of 10 K/min. A minimum of three samples per material were tested for a reasonable standard deviation.

### 2.5. Dynamic Mechanical Analysis

The dynamic mechanical properties of the carbon fiber-reinforced composites were measured according to DIN EN ISO 6721-1 with Rheometrics RDA-III (RHEO Service GmbH & Co. KG, Reichelsheim, Germany) with a heat ramp of 3 K/min, frequency of 1 Hz, and strain of 0.1% with a sample size of 50 × 10 × 2 mm^3^.

### 2.6. Compressive Testing of Epoxy Resins

Compressive tests of the neat resin were performed at room temperature according to DIN EN ISO 604 with a universal testing machine 1475 (ZwickRoell GmbH & Co. KG, Ulm, Germany). Samples with a length of 50 mm were used to evaluate the modulus, samples of 10 × 10 × 4 mm to evaluate mechanical strength.

### 2.7. Celanese Compressive Testing of Carbon Fiber-Reinforced Composites

Celanese Compressive tests were carried out according to DIN EN ISO 14126 with a universal testing machine 1475 (ZwickRoell GmbH & Co. KG, Ulm, Germany) with type A sample geometry. Figure 1 shows the test setup and sample preparation. A test speed of 1 mm/min was set according to the standard. Before the start of the test procedure, a waiting period of seven minutes was set to ensure homogenous temperature conditions in the climate chamber and through the composite sample. The temperature conditions were measured in pretests. As briefly described before, the main challenge of testing endless-fiber reinforced compositesat high temperatures up to 250 °C is to be able to conduct the force as shear from the Celanese set-up to the sample itself.

## 3. Results and Discussions

This section presents and discusses thermal, thermo-mechanical and mechanical properties of the carbon fiber-reinforced laminates together with the non-destructive ultrasound scan testing.

### 3.1. Thermal Analysis of the Neat Epoxy Systems

To be able to understand the thermal stability of the epoxy resins cured with the aliphatic (TETA), cycloaliphatic (IPDA), and aromatic hardener (DDS), the thermo-gravimetric analysis is applied to the fully-cured resin systems from 23 °C up to 250 °C. Figure 2 shows the weight loss of the composites versus temperature under air atmosphere.

The T_1%_ (temperature at which the weight loss of 1 wt% occurs) of DGEBA-TETA system was determined as 219 °C, for DGEBA-IPDA around 228 °C, and finally for DGEBA-DDS above 250 °C. The T_5%_ is above 250 °C for all samples. A slight mass loss with increasing temperature was observed for all types of resin systems but to a different extent in accordance with the higher stability of the aromatic hardener structure. TETA-cured epoxy resin showed the highest mass loss compared to the other resins. As the mass loss is very low, it should be attributed to the loss of water and volatile chemical groups on the surface of the samples.

### 3.2. Determination of Network Density

The network densities can be calculated theoretically or by simulation [9,10], by evaluation of the DMTA, as shown by Koleske [11], or by positron annihilation studies [12,13].

According to Murayama, the molecular weight between crosslinks, which can be viewed as the reciprocal value of the network density, can be calculated as follows: [9]
(1)$$M_{C}=\frac{M_{R}*EEW_{R}+M_{H}*EEW_{H}}{EEW_{R}+EEW_{H}}$$
where *EEW* is the epoxy equivalent weight of resin and hardener, and $M_{R}$ and $M_{H}$ are the molecular weights of resin and hardener. From this, the molecular weight between crosslinks is calculated as 142, 138, and 134 g/mol for TETA, IPDA and DDS, respectively. According to this theory, the network density is lowest for the system with TETA and highest for the one with DDS. Murayama also suggests calculating the modulus at the rubbery plateau G*^I^* by
(2)$$G^{I}=\rho *R*\frac{T}{M_{c}}$$
where ρ is the density, *R* the gas constant, and *T* the absolute temperature.

To deepen the understanding regarding the thermo-mechanical behavior of these polymers, the network densities have been calculated by means of the following equation, suggested by Koleske [14]:(3)$$\vartheta =\frac{G_{R}^{I}}{3\;R\;T}$$
where $G_{R}^{I}$ is the storage modulus at T, R the gas constant R = 8.3144598 J/(mol K), and T the absolute temperature 40 °C above the maximum of the loss tangent (T_g_) in Kelvin, where the rubbery plateau can be found. The rubbery plateau has been determined as stated in the publications mentioned.

The dynamic mechanical analysis of the RTM laminates was then performed to analyze the glass transition temperatures (T_g_). Values for T_g_ of the laminates were determined via the maximum of loss tangent and from the onset of the storage modulus G*^I^*.

Figure 3a shows the storage modulus of the laminates versus temperature. The onset started at 58 °C for the laminate with TETA, 105 °C for IPDA, and 148 °C for DDS. The storage modulus and thereby the loss tangent depended on the chemical structure of the matrix. The decrease of the storage modulus is more pronounced for the system with TETA than for IPDA and DDS. TETA is an aliphatic, IPDA a cyclo-aliphatic and DDS an aromatic hardener. TETA owns the weakest chemical bonds, the C–C single bonds have a bond energy of 348 kJ/mol, whereas the C=C double bonds in the aromatic ring of DDS own a bond energy of 602 kJ/mol. Furthermore, the phenyl ring in DDS leads to π–π stacking and the stacks have a bond energy of up to 50 kJ per mol [15].

Baranek et al. already showed the dependency of the length of the aliphatic bonds and the flexibility and T_g_ of aliphatic-bridged bisphenol-based polybenzoxazines [4,16]. Nakka et al. compared the thermo-mechanical properties by DMTA measurements of bisphenol A diglycidyl ether and four aliphatic amine curing agents with different chain length and could thereby show the flexibility of thermoset networks with aliphatic hardener [17]. As mentioned above, the aromatic nature of DDS leads to a polymer network with relatively low sensitivity to temperature, which has already been shown by Babayevsky et al., who analyzed the dynamic-mechanic behavior of DGEBA networks cured by DDS and 4,4′-Diaminodiphenylmethane (DDM) [14].

So far, networks based on aliphatic, cyclo-aliphatic, and aromatic hardeners have not been compared. Besides the dissipation energy mentioned above, the network density of the resulting thermoset networks influences their mechanical behavior [18]. With the equations presented in the theory section, the densities measured by the Archimedes principle and the resulting G*^I^* are shown in Table 3.

Clearly, these theoretical considerations do not take the topology of the networks into account [10]. To deepen the understanding regarding the thermo-mechanical behavior of these polymers, the network densities were calculated using the equations from Koleske stated in the theory part. Figure 3b highlights the loss tangent values measured by the dynamic mechanical thermal analysis (DMTA). The maximum of loss tangent was measured for the TETA-cured epoxy system at 101 °C, for IPDA at 138 °C and for DDS at 183 °C.

The network densities of the resin systems are shown in Table 4. The network densities increased from 559 mol/g for the system with TETA to 1069 mol/g for IPDA, and finally 1152 mol/g for the epoxy system where DDS hardener was used. Furthermore, the difference between storage modulus and loss modulus can be taken as a hint to show differences in network densities, as shown by Gottro [19]. As shown in Table 4, the difference between G’ and G’’ increases, which shows the increase in network density from TETA to IPDA and DDS.

When the theoretically calculated $G^{I}$ from Table 3 is compared to the measured $G_{R}^{I}$ from the DMTA, it can be clearly seen that the theory suggested by Murayama overestimates the modulus at the rubbery plateau. The differences can be attributed to neglecting the nature of the network structure in the theoretical model.

### 3.3. Compressive Test of the Neat Resin Systems

The compressive modulus of the resin systems was expected to have a strong effect on the compressive strength of the fiber-reinforced composites. This is because the tested laminates will only undergo very little straining until the failure of the material [20]. Therefore, the compressive behavior of the cured resin systems with aliphatic TETA, cycloaliphatic IPDA, and aromatic DDS were tested at 23 °C prior to compressive testing of their fiber-reinforced composites. The results are listed in Table 5.

As it can be observed from the testing results that the resin system cured with aliphatic TETA resulted in the lowest compressive strength at room temperature with 2.85 GPa compressive modulus. Following, the IPDA-cured epoxy resin showed a 28% higher compressive strength compared to the TETA-cured system but with a relatively lower compressive modulus of 2.74 GPa. Finally, the resin system cured with DDS hardener resulted in a very high compressive strength with a compressive modulus comparable to the IPDA system, which was measured as 2.79 GPa. The values are in accordance with the findings of other researchers [9]. The results seem valid, as the modulus is dependent on G’. The G’ of the laminates at room temperature was the highest for TETA with 3.55 GPa, compared to 3.06 GPa for IPDA, and 3.35 GPa for DDS (see Figure 3).

### 3.4. Ultrasound Testing and Optical Microscopy Measurements of the Laminates

To check the laminate quality after VARTM production, optical microscopy observations and ultrasonic sound testing of laminates were performed.

In Figure 4, the flaw echoes of the biaxial fiber-reinforced composites are shown. As can be seen from the flaw echoes, a very homogenous laminate quality without any observable micron-sized defects can be observed in the laminates with 2 mm laminate thickness. Only in the center of the laminates, the vacuum point can be observed, which is in general a resin overshoot and not defined as the defect. In general, C-scans show a very comparable laminate quality for all laminates.

Since the ultrasound scan has a minimum resolution of 5 microns, additionally, the through-thickness laminate quality was further analyzed via optical microscopy. The micrographs are shown in Figure 5. The fiber bundles of 0 and 90° for all types of laminates are clearly observable. Microstructures of all types of laminates seem to be comparable and no laminate defects or air bubbles were observed.

Finally, the fiber volume content of the laminates was characterized via thermo-gravimetric analysis. The fiber volume contents measured were 50, 49, and 47 vol% for TETA-, IPDA-, and DDS-cured laminates, respectively.

Consequently, laminate quality and fiber volume contents are highly comparable for each type of laminate, allowing for the possibility of characterizing the effect of the hardener and network density on the compressive behavior of the composites.

### 3.5. Celanese Compressive Test of the Fiber-Reinforced Composites at High Temperatures

In the following, the compressive behavior of the biaxial carbon fiber epoxy laminates at temperatures up to 250 °C is presented and discussed. In Figure 6, the compressive strength results are shown. The horizontal lines show the maximum of the loss tangent.

Figure 6 shows the effect of temperature on the compressive strength of the laminates in absolute values. The values were normalized for fiber volume content, which was highly comparable with 50, 49, and 47 vol% for TETA, IPDA, and DDS laminates, respectively (calculated via density). For the laminates cured with TETA and IPDA, no valid compression strength could be determined at 250 °C, which is why these values are missing in the graph. At room temperature, the values varied between 541 MPa for the laminates based on IPDA and 620 MPa for those based on DDS, only slightly higher than the value of 614 MPa for TETA. The values seem valid, as the compressive strength is dependent on the modulus of the resin. As presented above, the modulus was the highest for the resin systems with TETA and lowest for those with IPDA. The G*^I^* of the laminates at room temperature for TETA was the highest with 3.55 GPa, compared to 3.06 GPa for IPDA and 3.35 GPa for DDS. The modulus of the resin is crucial for the compressive strength of the laminate, as the strain at break is small for carbon fiber-reinforced laminates. In our tests, the strain at break was clearly below 1.5% up to the maximum of the loss tangent.

For TETA, a sudden decrease of the compressive strength from 614 MPa to 386 MPa (−47%) can be found at 110 °C, whereas the laminates based on IPDA and DDS forfeit their strength at higher temperature levels. The remaining mechanical properties can be explained by the network densities and the chemical structure of the thermoset network resulting from the different hardeners. As mentioned above, TETA leads to the lowest and DDS to the highest network densities at the rubbery plateau. Therefore, networks with TETA show the reduction of strength at much lower temperatures than IPDA and DDS.

This is also visible in Figure 7, which shows the normalized compression strength relative to the strength at room temperature. It should be mentioned that at higher temperatures (>180 °C), the differences between the samples based on TETA and IPDA were marginal. As the remaining mechanical strength at these temperatures was at a very low level, the fibers and the test setup of the Celanese compressive test may play an important role. The remaining properties at higher temperatures are therefore based on the textile and not on the epoxy matrix of the samples. DDS showed a slightly higher remaining compressive strength above the maximum of the loss tangent, which may be attributed to higher G’ at the plateau after the decrease of the compressive strength.

For a better understanding of the trend of the compressive strength, the values were fitted using a semi-empirical equation suggested by Feih [21]. The remaining compressive strength can be calculated by means of
(4)$$\sigma_{c}\left(T\right)=\frac{\sigma_{c\left(0\right)}+\sigma_{c\left(R\right)}}{2}-\frac{\sigma_{c\left(0\right)}-\sigma_{c\left(R\right)}}{2}\;\tan h\left(\phi \left(T-T_{K}\right)\right)$$
where $\sigma_{c\left(0\right)}$ is the compressive strength at the plateau before the decrease and $\sigma_{c\left(R\right)}$ after decrease. $T_{K}$ is the temperature where 50% of the mechanical properties remain, and ϕ is a material constant which needs to be fitted. The calculated values can be found in Table 6. Figure 6 shows the fitted curves with the calculated values. For TETA and IPDA, R^2^ is ~99%, which proves appropriate fittings could be found. DDS assumes an R^2^ of 96.9%.

It can be seen from Figure 7 that the value for DDS at a lower temperature of 110 °C is slightly overestimated by the equation, while the remaining strength at 185 °C is underestimated. For DDS, the retention of strength at higher temperatures of 225 and 250 °C was noticeable. While TETA and IPDA showed a sudden decrease of the mechanical properties at the maximum of loss tangent, a slightly higher strength was obtained for the laminates based on DDS.

The analysis of the network densities showed significant differences between TETA and IPDA, while the differences between IPDA and DDS were much smaller (1069 vs. 1152 mol/mm3). This is also partly reflected in the Tk values, where differences between TETA and IPDA (28 °C) were more significant than those between IPDA and DDS (10 °C).

There were differences in the fitting parameter ϕ, which cannot be further evaluated in this study. The parameter represents different influences, such as the effect of the fiber. The differences for ϕ between TETA and DDS were much smaller than between TETA and IPDA.

Furthermore, the failure modes of the laminates were analyzed. Robinson presented typical failure mechanisms in polymer matrix composites, which were evaluated [7]. Figure 8 shows optical photographs of the destroyed samples, tested at room temperature, 80 °C, and 225 °C (from top to bottom). It can be seen that all tests are valid, as the failure occurred between the clamping. For all samples, a kink band formation seems to be the reason for the failure in compressive tests. They usually start at the point of highest stress and continue in a specific angle to the other side of the sample. From the photographs, no significant differences between the samples based on TETA, IPDA, and DDS could be found.

## 4. Conclusions

Compressive behavior is a very important property for the construction of fiber-reinforced polymers. The evaluation of the influence of elevated temperatures on the compressive behavior of fiber-reinforced composites is therefore crucial for many applications. From the presented work, the following conclusions can be drawn:The differences in network densities calculated from theoretical considerations by Murayama do not seem to be as significant as those calculated from DMTA. This can be attributed to the neglect of the chemical structure and topology in the model by Murayama.The compressive strength of the laminates at room temperature is dependent on the modulus of the resin. A higher compressive modulus resulted in higher compressive strength in the Celanese compressive test. This could be explained by the fact that the laminate samples underwent a rather low deformation.A high correlation could be found between the thermo-mechanical properties measured in torsion by the DMTA and the scope of the compressive strength in the Celanese compressive tests. The compressive behavior of the composites could be fitted well with the equations suggested by Feih (R^2^ > 97%).

It can be concluded that in general, higher network densities of the resin matrix lead to a better mechanical performance in compressive tests at elevated temperatures. This might be an important structure–property relationship, which can be used in simulations or the construction with fiber-reinforced thermoset materials. It is suggested to further evaluate the properties of the fitted parameter ϕ, which might depend on the fiber orientation, properties of the fiber, and the mechanical properties of the resin.

## Figures and Tables

**Figure 1 polymers-11-00943-f001:**
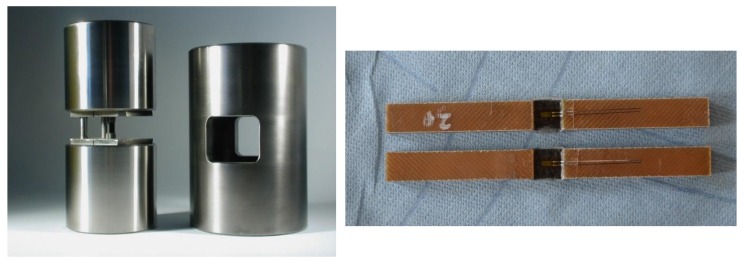
Pictures of Celanese test setup and sample preparation.

**Figure 2 polymers-11-00943-f002:**
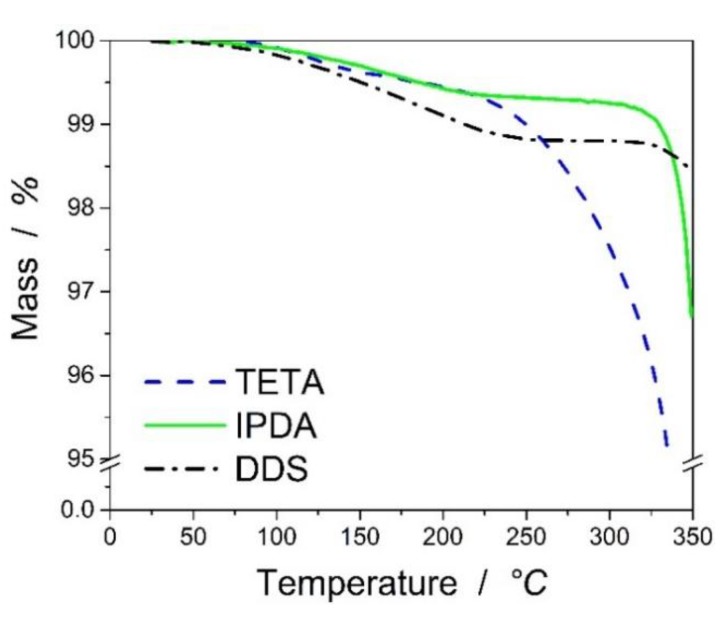
TGA of the resin under air atmosphere at 10 K/min heating rate.

**Figure 3 polymers-11-00943-f003:**
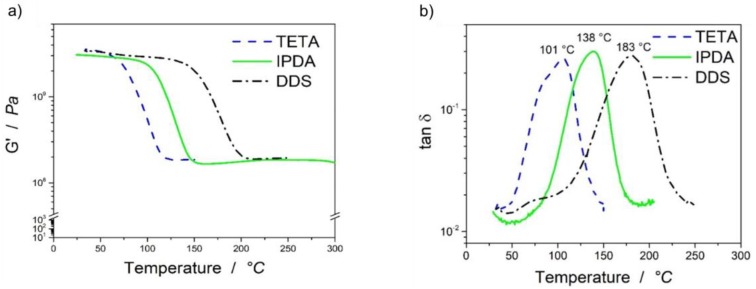
(**a**) Storage modulus (*G^I^*) and (**b**) tan(δ) of the resin transfer molding (RTM) laminates cured with various hardeners.

**Figure 4 polymers-11-00943-f004:**
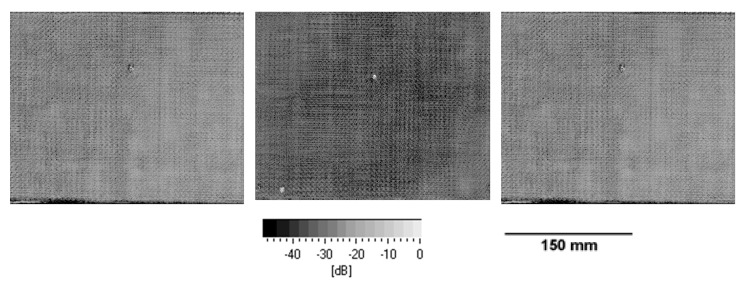
C-scans (flaw-echo) of the laminates (2 mm laminate thickness) cured with hardeners TETA, IPDA, and DDS (left to right).

**Figure 5 polymers-11-00943-f005:**
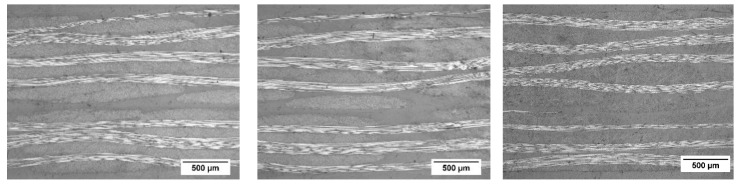
Optical microscopy of the laminates with TETA, IPDA, and DDS (left to right).

**Figure 6 polymers-11-00943-f006:**
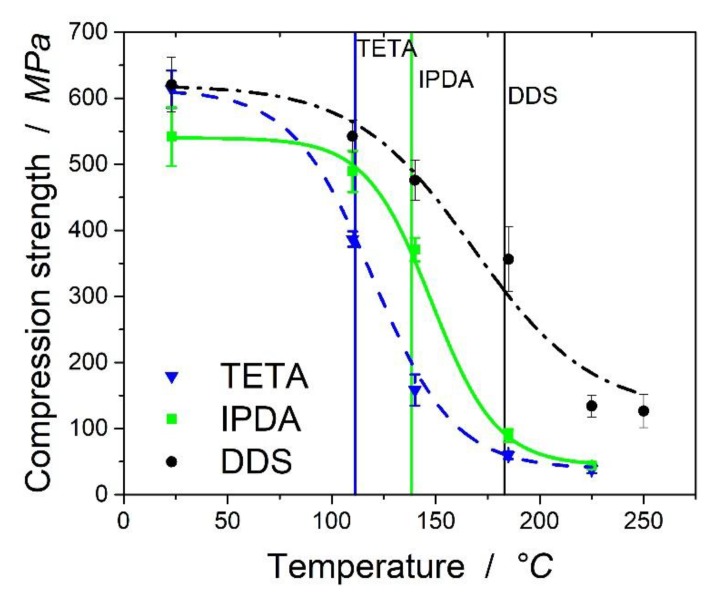
Compressive strength of the biaxial carbon fiber epoxy laminates with TETA, DDS, and IPDA hardeners. Fitting was performed with the formula suggested by Feih.

**Figure 7 polymers-11-00943-f007:**
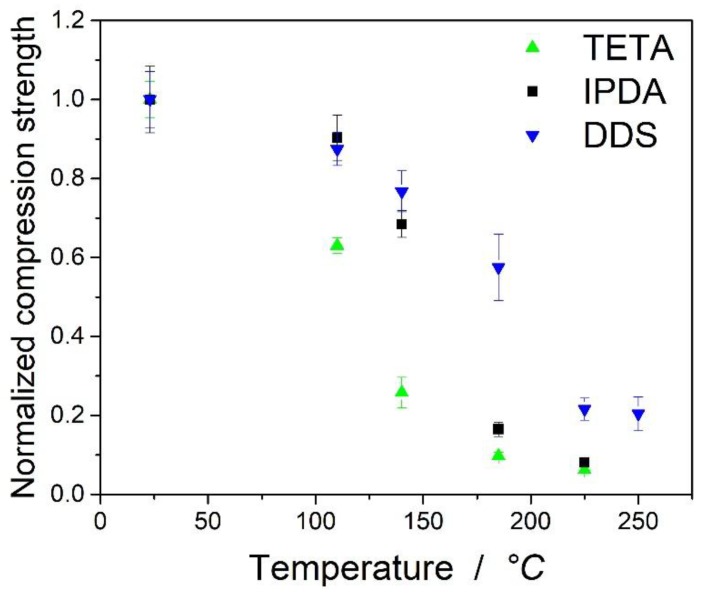
Normalized compression strength relative to the strength at room temperature.

**Figure 8 polymers-11-00943-f008:**
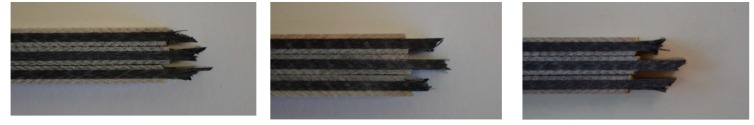
Fracture pattern of samples cured with TETA (**left**), IPDA (**middle**) and DDS (**right**).

**Table 1 polymers-11-00943-t001:** Chemical properties of hardener used to prepare the samples and laminates.

Hardener Name	Chemical Structure	Molecular Weight (g mol^−1^)	AHEW g/eq
**Triethylentetramine (TETA)**	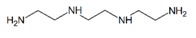	146.24	37.75
**Isophoronediamine (IPDA)**	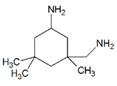	170.30	42.58
**4,4‘-Diaminodiphenylsulfon (DDS)**	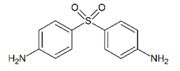	248.31	62.08

**Table 2 polymers-11-00943-t002:** Curing cycle for produced laminates.

Hardener	Step 1	Step 2	Step 3
TETA	10 min, 80 °C	10 min, 100 °C	50 min, 120 °C
IPDA	60 min, 80 °C	30 min, 120 °C	30 min, 160 °C
DDS	60 min, 140 °C	120 min, 180 °C	60 min, 200 °C

**Table 3 polymers-11-00943-t003:** Calculated values obtained by the Equations (1) and (2).

Hardener	$M_{C}\;g/\mathrm{mol}$	ρ g/cm^3^	$G^{I}\;\mathrm{Calculated}\;\mathrm{MPa}$
TETA	142	1.19	20.41
IPDA	138	1.14	20.02
DDS	134	1.23	22.33

**Table 4 polymers-11-00943-t004:** Calculated network densities for the matrix polymers.

Hardener	$G_{R}^{I}$ MPa	T °C	ϑ mol/g	*G^II^* KPa	Δ (*G^I^*, *G^II^*) MPa
TETA	5.756	141	559	3.393	5.75
IPDA	12.076	178	1069	675.5	11.4
DDS	14.169	223	1152	229.9	13.4

**Table 5 polymers-11-00943-t005:** Results from compressive tests at room temperature of resin with different hardeners.

Hardener Type	Compressive Strength MPa	Compressive Modulus GPa
TETA	148 ± 8	2.85 ± 0.28
IPDA	204 ± 17	2.74 ± 0.25
DDS	297 ± 20	2.79 ± 0.30

**Table 6 polymers-11-00943-t006:** Parameters calculated for the fitting function suggested by Feih.

Hardener	$\sigma_{c\left(0\right)}$	$\sigma_{c\left(R\right)}$	$T_{K}$	ϕ	R^2^
	MPa	MPa	°C		
TETA	613.6	38.6	120	0.025	0.997
IPDA	541.2	43.4	148	0.031	0.998
DDS	620.0	126.1	168	0.017	0.969

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
