# Peer review of "Effect of Curing Agent on the Compressive Behavior at Elevated Test Temperature of Carbon Fiber-Reinforced Epoxy Composites"

_polymers, 2019, doi:10.3390/polym11060943_

Round 1

Reviewer 1 Report

The paper titled “Effect of curing agent on the compressive behavior at elevated test temperature of carbon fiber reinforced epoxy composites” are dealing with the change of thermal properties of the composites based on different hardeners.

I think that it is good to share with the readers of Polymers. However, there are some points to be corrected.

Its structure would be 1. Introduction, 2. Experiments, 3. Discussion, 4. Conclusion

There are some typos and grammatical errors like isophoron (isophorone is correct though)

I cannot still understand what is end-taps made of epoxy and woven glass fiber, and glued together. There should be more detailed explanation for the preparation of test samples. Furthermore, I think that there should be brief explanation about the glue such as components, curing conditions, and why they used it instead of commercial one since the author mentioned it as own developed one.

 For figure 2, and table 3, I do not understand why the authors show T1% and T5% instead of thermal degradation temperature of each sample in TGA experiments, and why they empathize T1% in table 3. The reason should be explained enough.

Some figures used left and right for each figure. It may be appropriate to use (a), and (b)

Author Response

All changes were made as requested. A detailed response can be found attached.

Reviewer 2 Report

There are several minor points that need to be addressed on the experimental side

1) What is the weight ratio or molar ratios of the components used to prepare the samples? These needs to be clarified in the experimental section. 

2) Can the authors show that the curing cycle that they used have fully consumed the epoxy groups?

3) Does the secondary amine in the TETA participate in the crosslinking reaction? I am assuming it will and I'm a little surprised that it led to the lowest crosslink density. Again, I don't know what I'm supposed to expect if I don't know how much crosslinker and resins were being mixed together. 

Author Response

(The authors gave the same response as above.)
